# Enhanced Insecticidal Effect and Interface Behavior of Nicotine Hydrochloride Solution by a Vesicle Surfactant

**DOI:** 10.3390/molecules27206916

**Published:** 2022-10-15

**Authors:** Wenjun Xiao, Xiufang Cao, Pengji Yao, Vasil M. Garamus, Qibin Chen, Jiagao Cheng, Aihua Zou

**Affiliations:** 1School of Chemistry and Molecular Engineering, East China University of Science and Technology, Shanghai 200237, China; 2College of Science, Huazhong Agricultural University, Wuhan 430070, China; 3Department of Power-Based Materials Development, Institute of Metallic Biomaterials, Helmholtz-Zentrum Hereon, Max-Planck-Straße 1, 21502 Geesthacht, Germany; 4Shanghai Key Laboratory of Chemical Biology, School of Pharmacy, East China University of Science and Technology, Shanghai 200237, China; 5Shanghai Key Laboratory of Rare Earth Functional Materials, College of Chemistry and Materials Science, Shanghai Normal University, Shanghai 200234, China

**Keywords:** nicotine hydrochloride, vesicles, surfactant, insecticide, interface property

## Abstract

Nicotine hydrochloride (NCT) has a good control effect on hemiptera pests, but its poor interfacial behavior on the hydrophobic leaf leads to few practical applications. In this study, a vesicle solution by the eco-friendly surfactant, sodium diisooctyl succinate sulfonate (AOT), was prepared as the pesticide carrier for NCT. The physical chemical properties of NCT-loaded AOT vesicles (NCT/AOT) were investigated by techniques such as dynamic light scattering (DLS), small-angle X-ray scattering (SAXS), and cryogenic transmission electron microscopy (cryo-TEM). The results showed that the pesticide loading and encapsulation efficiency of NCT/AOT were 10.6% and 94.8%, respectively. The size of NCT/AOT vesicle was about 177 nm. SAXS and surface tension results indicated that the structure of the NCT/AOT vesicle still existed with low surface tension even after being diluted 200 times. The contact angle of NCT/AOT was always below 30°, which means it could wet the surface of the cabbage leaf well. Consequently, NCT/AOT vesicles could effectively reduce the bounce of pesticide droplets. In vitro release experiments showed that NCT/AOT vesicles had sustained release properties; 60% of NCT in NCT/AOT released after 24 h, and 80% after 48 h. Insecticidal activity assays against aphids revealed that AOT vesicles exhibited insecticidal activity and could have a synergistic insecticidal effect with NCT after the loading of NCT. Thus, the NCT/AOT vesicles significantly improved the insecticidal efficiency of NCT, which has potential application in agricultural production activities.

## 1. Introduction

Pesticides play a major role in protecting crops and promoting the increase in agricultural production. Nearly 80% of fruit production and half of vegetable production would be lost without the use of pesticides [1]. The most widely used pesticides in China are synthesized by chemical methods, which can cause Colony Collapse Disorder (CCD) and also do great harm to birds [2,3,4]. Moreover, most of the pesticides is lost during spraying, causing great pollution to the environment, especially to water [5,6]. Zhang detected the presence of a variety of nicotinic insecticides in the Pearl River and Xiangjiang River basins, and the biological risk assessment results showed that the pesticides had caused certain damage to the ecology of the local invertebrate organisms [7].

Biogenic pesticides have been widely studied because of their good bioselectivity, degradability, and low drug resistance [8,9,10]. Nicotine is an alkaloid extracted from plants in the nightshade family (mainly tobacco) and has good stomach toxicity effects and biodegradability [11,12,13]. However, the spreading performance of nicotine on the leaf surface is poor, making it difficult to use in practice. Nanotechnology, which has emerged in recent years, provides a better solution [14,15]. The commonly used nanoparticles mainly include nanomicelles [16], nanogels [17], nanocapsules [18], nanoliposomes [19], and porous inorganic nanomaterials [20]. Nanoaminic preparations can provide pesticides with a variety of properties and behavioral improvements, such as solubility, dispersity, stability, mobility, targeted delivery, runoff, and environmental residues [21,22].

Different formulations of nicotine nano-pesticides have been studied, including nicotine oleate formulations [23], nicotine carboxylate emulsions [24], and polymeric nanoparticles [25]. Nicotine oleate formulations mainly improved the biological activity and reduced the viscosity and dispersion time of nicotine. Nicotine carboxylate emulsions were mainly studied in the context of the influence of different lengths of fatty acid chains on the encapsulation efficiency and insecticidal efficiency of the preparation. Our group prepared NTC hydrochloride nanoparticles encapsulated by CS/TPP in previous research, and we studied the influence of the presence of monovalent salts on the properties of the nanoparticles [26]. The above formulations can protect nicotine from the external environment, but the various behaviors on the leaves such as bounce retention time have not been studied. Studies have shown that more than 50% of agrochemicals will pollute the soil and water bodies due to the behavior of bouncing and splashing [27,28].

AOT is an anionic surfactant with double hydrophobic chains, which can spontaneously form vesicles when its concentration reaches 1 × 10^−3^ mol·L^−1^ [29,30]. AOT is an excellent emulsifier, detergent, and penetrant agent used in the textile industry with excellent permeability and wettability. According to the Material Safety Data Sheet (MSDS) of AOT, the acute oral toxicity LD_50_ of AOT to rats is larger than 3000 mg/kg, and the dermal LD_50_ to rabbits is larger than 10,000 mg/kg, which means AOT has good safety. Jiang compared the micellar solution formed by sodium lauryl sulfate and trisiloxane to the AOT vesicle solution for surface tension, droplet bounce, and contact angle [31]. With the same concentration, AOT vesicles were significantly better than the micellar solution formed by sodium lauryl sulfate and trisiloxane in enhancing the spread of solution on the leaf surface and inhibiting the bounce of droplets.

In this study, an NCT/AOT vesicle solution was prepared in order to improve the interface properties of the nicotine on the target crops and enhance its target deposition performance. The characterization of wettability, anti-fog drop bounce performance, and insecticidal activity of NCT/AOT vesicle solution were studied. The effect of AOT vesicle solution on improving the utilization of NCT was evaluated.

## 2. Results

### 2.1. The Preparation of NCT/AOT Vesicles

The anionic surfactant AOT itself can form a vesicle structure at certain concentrations [32]. The same concentration (15 mM) of NCT hydrochloric acid solution and AOT aqueous solution were mixed at different molar ratios in order to find the optimal molar ratio for the formation of NCT/AOT vesicles. The molar ratio of NCT/AOT had a significant impact on the phase behavior of the NCT/AOT aqueous system (Figure 1A). The NCT/AOT samples of the molar ratios of 2/10, 3/10, and 4/10 showed a light blue-colored state, which indicates that a vesicle structure may have formed in the system, as shown in Figure 1B. The NCT/AOT samples of the molar ratios of 5/10 and 6/10 appeared milky with floccules, which indicated that precipitation appeared due to the neutralization of opposite charges of AOT and NCT molecules. The NCT/AOT samples of the molar ratios of 7/10, 8/10, and 9/10 showed obvious aggregation of precipitation, and oil droplets appeared after standing for 5 h, which indicates that the stability of the system was poor. The optimal molar ratio of NCT/AOT of 3/10 was selected for subsequent experiments.

The encapsulation efficiency and pesticide loading of NCT/AOT samples were measured by the ultrafiltration/centrifugation method. The concentration of NCT (free NCT concentration) in the centrifugal fluid was determined after ultrafiltration. According to Equations (1) and (2) and Appendix A, it could be calculated that the encapsulation rate of NCT/AOT vesicles was 94.8%, and the pesticide loading was 10.6%. The encapsulation rate was much higher than that of nicotine nanoparticles (84.7%) and nicotine emulsions (54.1%), as reported in previous work [24,26].
(1)$$EE\%=\frac{C_{T}-C_{U}}{C_{T}}\times 100\%$$
(2)$$PL\%=\frac{(C_{T}-C_{U})\times V}{(C_{T}-C_{U})\times V+m_{AOT}}\times 100\%$$
where *C_U_* indicates the concentration of the pesticide that is not included; *C_T_* indicates the total concentration of the added pesticide; *V* represents the total volume added; and *m_AOT_* represents the total amount of AOT added.

NaCl salt in the NCT/AOT preparation was removed according to a previous method in order to prevent its influence on plant growth [33]. Briefly, the mix solution of AOT and NCT was freeze-dried and dissolved in ethanol to precipitate NaCl. Then it was centrifuged at 5000 rpm for 5 min. The final NCT/AOT was obtained and stored after the supernatant was totally dried in a vacuum drying oven. The resulting solid product was redissolved in water and subjected to all subsequent tests.

### 2.2. Structure Characterization of NCT/AOT Vesicles

DLS is one of the conventional methods to characterize colloidal nanoparticles. The cumulant method was used for analysis, and intensity data were presented. The values of particle size, PDI, and Zeta potential for NCT/AOT vesicles without dilution were 198 nm, 0.23, and −56.5 mV, respectively.

The NCT/AOT sample was then diluted by a certain multiple, and the particle size, PDI, and Zeta potential were also determined (Figure 2). The particle sizes of NCT/AOT diluted by different times was 187 nm (200 times), 186 nm (500 times), 178 nm (1000 times), and 159 nm (2000 times). For the PDI values, they were all less than 0.30, as shown in Appendix A. Figure 2 also showed that the Zeta potential value was reduced from −56.5 ± 2.3 mV to −32.8 ± 2.0 mV.

The SAXS data were fitted by the monodispersing approach of bilayer vesicles, and the results are shown in Figure 3A. There is an obvious peak in the intermediate q-vector range (q = 0.26), indicating the presence of a vesicle structure. The slope of the scattering intensities (α) in the lowest q-vector range became larger after the sample was diluted, which implied that the vesicle is more compact at higher concentrations. A simple bilayer vesicle graph was drawn based on the fitting parameters. R represents the radius of the vesicle; t_t represents the thickness of the inner part of the bilayer; and t_h represents the thickness of the outer part of the bilayer. The length of the AOT alcyl chain was about 0.8 nm. The tail chains overlapped partially in the process of vesicle formation, making t_t slightly less than 1.6 nm. A wide range of particle size distributions (30–200 nm) was obtained during the fitting process, consistent with DLS results.

Cryogenic transmission electron microscopy was further performed to identify the morphology of the NCT/AOT solution. Figure 3B illustrates that there were some unilamellar vesicles formed with sizes less than 100 nm. The size value determined by cryo-TEM was smaller than the results determined by DLS and SAXS, which is mainly because cryo-TEM measures samples after freezing, while DLS and SAXS measure their hydration radius.

### 2.3. Interfacial Properties

The surface tension of the NCT/AOT system under different dilution ratios was determined, as shown in Figure 4B. The surface tension values for the NCT/AOT system under the dilution times of 200–2000 were all less than 30.00 mN/m, which is lower than the critical surface tension of cabbage leaves (36.26–39.00 mN/m) [34]. Therefore, the NCT/AOT formulations can wet the cabbage leaves in theory.

Contact angle is an intuitive evaluation standard for pesticide formulations spreading on leaves and is used to judge the wetting of pesticide formulations and target crop leaves. This experiment simulated the static (Figure 4A) and dynamic contact angle (Figure 5) of the prepared NCT/AOT nano-pesticide formulation on cabbage leaves to evaluate its wetting performance. Three parallel experiments were performed for each sample. As shown in Figure 4, the average contact angle of pure water on the cabbage leaves was about 136° (greater than 135°), which indicates that the cabbage leaves had a super-hydrophobic structure and also that the prepared cabbage leaves preserved the original structure. The addition of AOT decreased the contact angle significantly, but the contact angle exceeded 60 after being diluted 1000 times, and the contact angle still increased with the increase of dilution rate. The contact angle of the NCT/AOT system on the cabbage leaf surface was always less than 30°, which proved the good wetting performance of NCT/AOT, even after dilution. It was also found that the contact angle of the NCT/AOT sample after being diluted 500 times was less than that of the sample diluted 200 times, mainly because the NCT/AOT sample after being diluted 200 times had the slightly higher viscosity [35].

The “droplet bouncing” behavior that occurs during the spraying process is also the main difficulty for traditional pesticide formulations. A study of the bouncing behavior of NCT/AOT nano-pesticides on the surface of cabbage leaves under different dilution multiples was performed to judge the pros and cons of the formulation in the actual application process. Figure 5 shows the dynamic process of drops of water and NCT/AOT vesicles on the leaf surface of cabbage. After being diluted 200, 500, 1000, or 2000 times, the droplets still could be evenly dispersed on the leaf surface after 1s of contact with the leaf surface, and the droplets did not shrink after 3 s and 5 s. In contrast, the contact angle of water on the leaf surface was always greater than 90 degrees, which means it could easily roll off the leaf surface.

The “droplet bouncing” behavior of water and NCT/AOT vesicle systems on the cabbage leaves are shown in Figure 6. When the droplet falls and hits the cabbage leaf surface, the surface of the droplet undergoes a large deformation. When the droplet spreads to the maximum diameter, due to the uneven force on the edge, the inward surface tension will re-polymerize the dispersed droplets, which will decompose into multiple small droplets after 1.2 ms [36,37]. Due to the special micro/nano structure of cabbage leaf, the capillary phenomenon of droplet spreading is produced. The vertical upward Laplace force will cause the droplets to “splash”, just like the phenomenon that occurs at 4.0 ms. Figure 6A shows that it took 47.0 ms for the water droplets to remain on the cabbage leaf in a completely non-wetting state. This result is consistent with the verification of the contact angle experiment, indicating that the intercepted cabbage leaf surface preserves the original structure and has super-hydrophobic properties. The behavior of NCT/AOT vesicle droplets hitting cabbage leaves at different dilutions times is shown in Figure 6B. Under the condition of 200-fold dilution times, the NCT/AOT vesicle system did not bounce and roll off, and it was completely wetted in 2.0 ms. This is mainly because NCT/AOT has a vesicle structure, which can be densely packed on the leaf surface of cabbage, thereby reducing the occurrence of the “capillary phenomenon”. Under the condition of 500-fold dilution, the droplet spread to the largest area first and then slightly retracted. After 4.0 ms, the droplets could spread out. Under the condition of 1000-fold dilution, the droplets rebounded to a large extent. The droplet first spread to the maximum radius. The micro-nano structure on the leaf surface was not tightly arranged. The upward Laplace force was generated where air exists, and the potential energy was converted into kinetic energy again. After several rounds, the droplet could still be spread evenly on the leaf surface after 13.6 ms, because the static surface tension of the droplet was smaller than that of the leaf surface [38,39].

### 2.4. Release Behavior

The release performance of NCT/AOT vesicles was studied by dialysis. Double-distilled water was used as the sustained-release medium, and the NCT solution was used as a control. As shown in Figure 7, the release of the pure NCT was very rapid. The cumulative release of NCT reached more than 80% after 0.5 h, and 100% after 1 h. This is mainly because the release of NCT is in full compliance with Fick’s first law. The greater the concentration gradient inside and outside the dialysis bag, the greater the diffusion flux. The NCT in the dialysis bag was more likely to diffuse into the external release medium. The NCT release from NCT/AOT vesicles did not have “burst release behavior” initially, and the cumulative release reached 60.0% after 24 h. After 48 h, the release basically reached equilibrium, and the total release reached more than 80.0%.

### 2.5. Insecticidal Activity

With the purpose of proving the applicability of NCT/AOT vesicles in agricultural production activities, assays of insecticidal activity of NCT/AOTagainst aphids were measured with NCT and AOT as controls. The virulence regression equation, LC_50_ value, and correlation coefficient are listed in Appendix A. As seen in Appendix A, it was determined that AOT itself has a certain insecticidal activity. The LC_50_ values of NCT, AOT, and AOT-NCT were 91.68, 99.03, and 10.55 ppm, respectively. The results show that the insecticidal activity of NCT/AOT is significantly higher than that of NCT. The encapsulation of NCT into AOT vesicles decreases LC_50_ roughly 10 times, since AOT prevents NCT from being affected by the external environment. Moreover, AOT and NCT can play a synergistic insecticidal role after NCT is loaded into AOT vesicles.

## 3. Discussion

In order to improve the efficiency of pesticides and reduce their pollution to the environment, a vesicle solution using an eco-friendly surfactant, AOT, was prepared as the pesticide carrier for NCT. When AOT was mixed with nicotine hydrochloride, the coexistence of Na^+^ and Cl^–^ decreased the hydrophilic surface area, resulting in the increase of the molecular packing parameters, and vesicles formed more easily [40]. The pKa of AOT is 5.75 [33,41], and as the ratio of nicotine hydrochloride was increased, the SO_3_^2−^ group of AOT bonded with H^+^, and bis (2-ethylhexyl) sulfosuccinic acid was formed [42]. Then oil-like droplets formed when the concentration of bis (2-ethylhexyl) sulfosuccinic acid reached a certain level. The pH of different samples is shown in Appendix A. The Zeta potential value of NCT/AOT was −56.1 mV, which was higher than −30 mV, even after 2000 times dilution, indicating excellent stability of the sample. It can also be seen that when the dilution ratio reached 2000 times, the PDI and particle size significantly decreased. AOT can form micelles at a certain concentration. With the dilution of NCT/AOT, some vesicles may transform into micelles and form spherical structures with smaller radii. Both SAXS and cryo-TEM results showed good spherical vesicle structures.

During the dilution process of NCT/AOT, it was found that the contact angle of the sample showed an increasing trend. However, the contact angle of NCT/AOT diluted 500 times was smaller than that of the sample diluted 200 times, mainly because the NCT/AOT sample after 200 times dilution had a slightly higher viscosity [35]. A contact angle less than 60° indicated that the wettability was good [43]. The contact angle of NCT/AOT was always below 30°, even after being diluted 2000 times, indicating that NCT/AOT nano-pesticides have good interface properties and can improve the wetting performance of pesticide formulations on the foliage of target crops in actual applications. Moreover, NCT/AOT could always be deposited on the leaf surface after 1s, which could reduce the loss of NCT. In the test of bounce performance, it can be concluded that the NCT/AOT vesicle structure can effectively reduce the bounce behavior of droplets hitting the leaf surface, improving the affinity with the leaf surface.

It takes a certain amount of time for NCT to pass through the bilayer to reach the external environment because of the bilayer structure of NCT/AOT vesicles, so NCT/AOT vesicles have a sustained release property and can reduce the loss amount of pesticides used. It was seen that the release rate at 36 h decreased significantly. After being released from vesicles, NCT entered the slow-release medium through the dialysis bag. This process was affected by both the electrostatic action of vesicles and the osmosis of the concentration gradient. In the later stage of release, the concentration difference between the two sides of the dialysis bag decreased, but the vesicle structure changed little, and the release rate slowed down accordingly. In summary, NCT could release slowly and continuously over 80 h, extending the duration of its effects. The insecticidal effect of NCT was significantly improved with the improvement of its sustained release and interface performance, leading to certain advantages in the practical application of NCT/AOT vesicles.

## 4. Materials and Methods

### 4.1. Materials

AOT (≥99%) was purchased from Adamas (Shanghai, China). Hydrochloric acid (36.5%, wt%) was purchased from Shanghai Titan Technology (Shanghai, China). Nicotine (≥99%) was obtained from Huazhong Agricultural University(Wuhan, China). Ethanol (≥95%) was purchased from Shanghai Titan Technology (Shanghai, China).

### 4.2. Preparation of NCT/AOT Vesicles

The preparation of NCT/AOT vesicles was based on the electrostatic attraction method, as described previously and shown in Appendix A [40]. Briefly, the same concentrations of NCT hydrochloric acid solution and AOT aqueous solution were mixed and freeze-dried (FD-1A-50, Boyikang Instrument Equipment, Beijing, China). Then the obtained system was centrifuged (KDC-140HR, Anhui USTC Scientific Instruments CO., LTD, Anhui, China) at 5000 r/min for 5 min after ethanol was added. The obtained supernatant was kept under vacuum drying (DZF-6020, Newair Equipment, Shanghai, China) for three days and stored for further tests.

### 4.3. Encapsulation Efficiency and Pesticide Loading

The encapsulation efficiency (EE%) and pesticide loading (PL%) of the samples was measured by the ultrafiltration/centrifugation method. Each sample (4 mL) was centrifuged for 30 min at 10,000 r/min to remove the unencapsulated pesticide at the temperature of 4 °C. The amount of NCT in the filtrate was measured by ultraviolet−visible (UV–vis) at 269 nm. The PL% and EE% was calculated as Equations (1) and (2).

### 4.4. Surface Tension

Surface tension was measured by the Wilhelmy plate method (BZY-2, Equity Instruments Shanghai). The standard surface tension value of the water under 293 K is 72.75 × 10^−3^ N/m, which was used to calibrate the accuracy of the automatic surface tension meter. A platinum sheet was burned red and allowed to cool naturally. Then it was fixed on the hook and connected to the balance, and 5 mL of each sample was added after the calibration instrument was measured using redistilled water. The platinum sheet had to be washed and burned red after each measurement was completed. Each sample was subjected to three experiments in parallel.

### 4.5. DLS Determination

The particle size, polydispersity index (PDI), and Zeta potential of NCT/AOT vesicles and its diluted solutions were measured by DLS (Litesizer 500, Anton Paar, Shanghai, China). In order to ensure the accuracy of the experimental data, each sample was subjected to three experiments in parallel.

### 4.6. Contact Angle

The contact angle was measured (JC200D1, Zhongchen Digital Technology Equipment, Shanghai, China) on cabbage leaves, which were cut into 0.5 cm × 5 cm strips. The leaves were glued on a long slide, which was then fixed on the contact angle measuring instrument. Then 3 μL of sample was dropped onto the flat surface of the blade. The contact angle θ was measured by slowly adjusting the button three times. A charge-coupled device (CCD) was used to take pictures after the droplets contacted the leaf surface from 0 s to 20 s.

### 4.7. Bounce Behavior

An ultra-high-speed camera (UX100 type 200K-M-16GB, Keyence, Osaka, Japan) was fixed on the bracket and was connected to a computer. The collision and bounce process of the fog droplets was captured after free fall. The shooting frequency was set at 8000 fps, and the frame number was 1280 × 616. The photos of the bounce part were manually filtered from collected videos.

### 4.8. SAXS Measurements

The internal structure of NCT/AOT vesicles was determined with the synchrotron radiation apparatus of the BL19U2 line of the Shanghai Synchrotron Radiation Source (Shanghai Ins titute of Applied Physics, Chinese Academy of Sciences, Shanghai, China). The distance between the sample and the scatterer was adjusted until the range of momentum transfer q (q = 4πsinθ/λ; 2θ is the scattering angle) was between 0.01 Å and 0.5 Å. The exposure duration was set to 1 s, and the X-ray wavelength was 1.033 Å. Each sample was measured 10 times, and the beam size passing through the center of the capillary was adjusted to 0.40 × 0.15 mm^2^ (H × V) during each measurement.

### 4.9. Cryo-TEM

Cryo-TEM was performed at the National Center for Protein Science, Shanghai (Tecnai G2 20, operating at 200 kv). A drop of AOT/NCT solution was spread on a TEM copper grid to form thin films suspended on the mesh holes. Then the copper grid was quickly placed into a container of liquid ethane cooled by liquid nitrogen (−165 °C). The vitrified sample was then stored in the liquid nitrogen until it was transferred into the vacuum column of the TEM microscope.

### 4.10. Release Behavior

Three milliliters of the prepared NCT/AOT vesicles and NCT solution was transferred into two dialysis bags (MWCO = 8–14 kDa, Yuanye Biotechnology Shanghai). Then the dialysis bags were placed in brown bottles containing 150 mL of slow-release medium and were oscillated slowly in a constant temperature oscillator at 25 °C. Then 3 mL of release medium was taken out at a certain time interval, and the same volume of fresh release medium was supplemented at the same time. The absorbance of nicotine release media at different times was measured by UV–vis. The concentration of NCT was calculated by the standard curve equation. The cumulative release amount (*Q*_n_) was calculated by Equation (3):(3)$$Q_{n}=C_{n}\times V_{0}+\overset{n-1}{\underset{i=1}{\sum }}C_{i}\times V_{i}$$
where *C_n_* represents the concentration of NCT in the sustained-release medium at each sampling point; *C_i_* represents the concentration of NCT in the sustained-release medium taken for the ith time; and *V_0_* and *V_i_* represent the total volume of redistilled water and the volume of the sample to be tested and taken out every time, respectively.

### 4.11. Insecticidal Activity Assays

The insecticidal activity of NCT/AOT, AOT, and NCT was tested using the following procedure [44]. Three adult aphids were inoculated on fresh bean sprouts and covered with a lampshade. After 16 h of inoculation, adult aphids were removed and checked to ensure that the number of aphids per plant was larger than 15. Three days after the inoculation, the broad bean stems and leaves (including aphids) were immersed in different concentrations of samples for 10 s, which were then dried, sealed, and covered with a lampshade. The experimental environmental conditions were set as follows: temperature 20.9–28.7 °C (average value 23.7 °C), humidity 40.6–75.0% (average value 58.2%), illumination 14 Lh:10 Dh, and 3 sets of experiments carried out in parallel.

## 5. Conclusions

In summary, an NCT-loaded vesicle (NCT/AOT) solution with high pesticide loading (10.6%) and encapsulation efficiency (94.8%) was successfully prepared by the ion exchange process. SAXS and cryo-TEM measurements both proved the formation of the vesicle structure. The size of NCT/AOT vesicles was about 170 nm and increased slightly after a certain multiple of dilution. The surface tension of NCT/AOT vesicles was 27 mN/m, and its contact angle on the leaf surface was below 30°, indicating that it could wet the leaf surface well. The droplet of NCT/AOT had no rebound under a low dilution ratio. All the results indicated that the interface behavior improved greatly after the formation of NCT/AOT vesicles. In addition, the NCT/AOT vesicle had a sustained release property, and NCT could release continuously over 80 h. An insecticidal experiment indicated that AOT and NCT play a synergistic insecticidal role, reducing the LC50 value by about 10 times, effectively improving the insecticidal ability of the NCT. This study highlighted the improvement of interface properties and the insecticidal ability of NCT after the formation of vesicles, which provides the possibility for the further application of NCT.

## Figures and Tables

**Figure 1 molecules-27-06916-f001:**
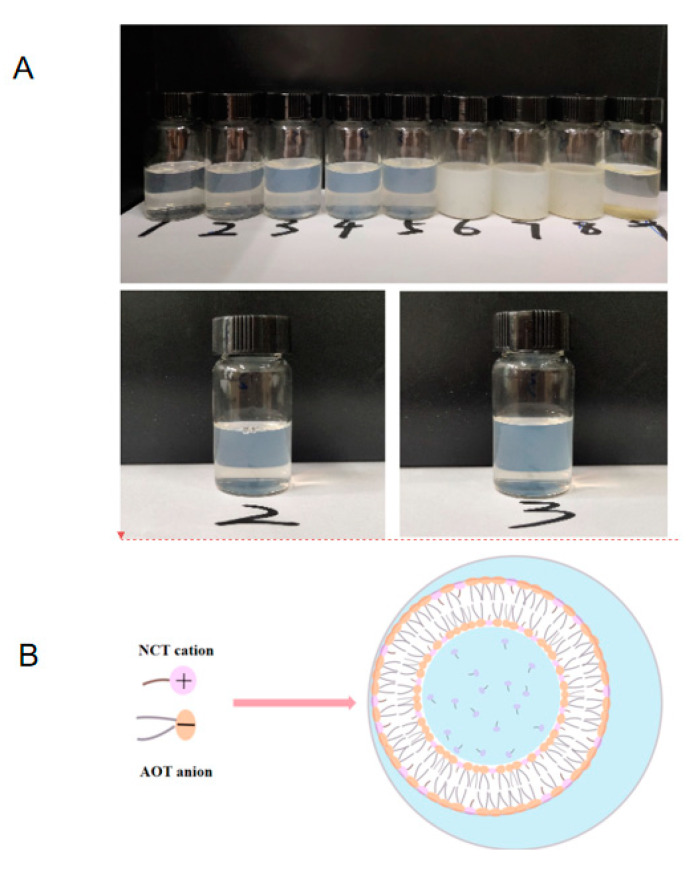
(**A**) The pictures of NCT/AOT samples at different molar ratios; 1, 2,…9 represents that the molar ratios of NCT and (AOT + NCT) are 1/10, 2/10…9/10. (**B**) The illustration of NCT/AOT vesicle formation.

**Figure 2 molecules-27-06916-f002:**
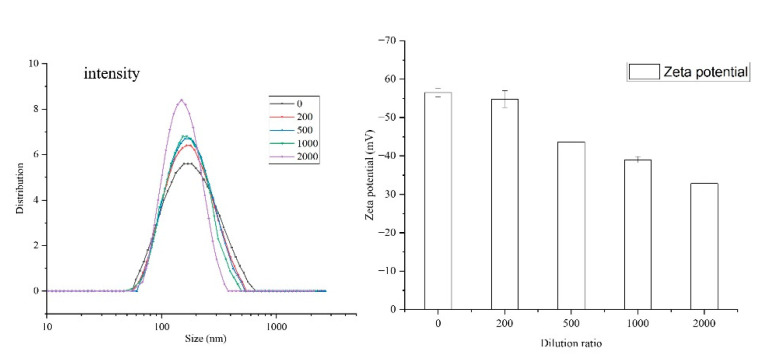
Particle size, PDI, and Zeta potential values of NCT/AOT with different dilution times.

**Figure 3 molecules-27-06916-f003:**
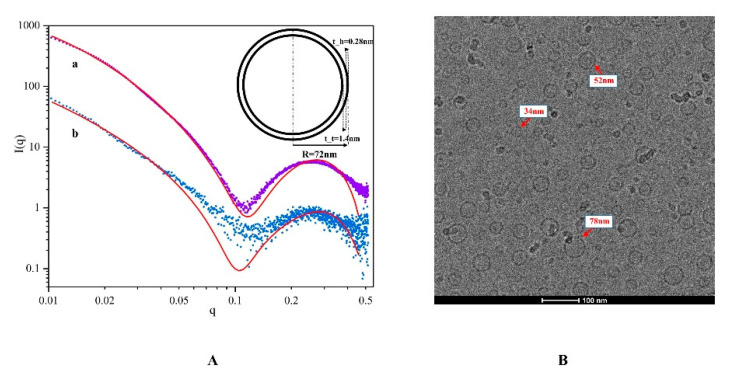
(**A**) Simulation of I(q)–q SAXS based on NCT/AOT (a) and in 200-fold dilution (b); (**B**) cryo-TEM image of NCT/AOT vesicles.

**Figure 4 molecules-27-06916-f004:**
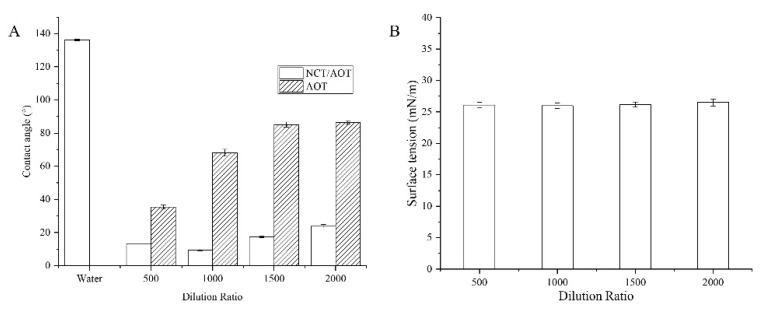
(**A**) Contact angle of NCT/AOT and AOT at different dilution ratios; (**B**) surface tension of NCT/AOT at different dilution ratio.

**Figure 5 molecules-27-06916-f005:**
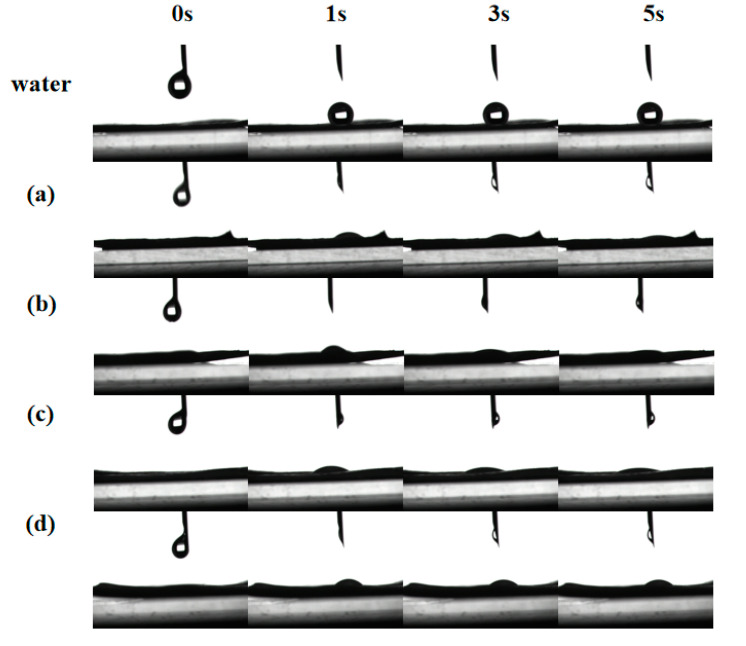
Dynamic contact angle of water and NCT/AOT droplets on the surface of cabbage leaf under different time with different dilution factors: diluted (**a**) 200 times, (**b**) 500 times, (**c**) 1000 times, and (**d**) 2000 times.

**Figure 6 molecules-27-06916-f006:**
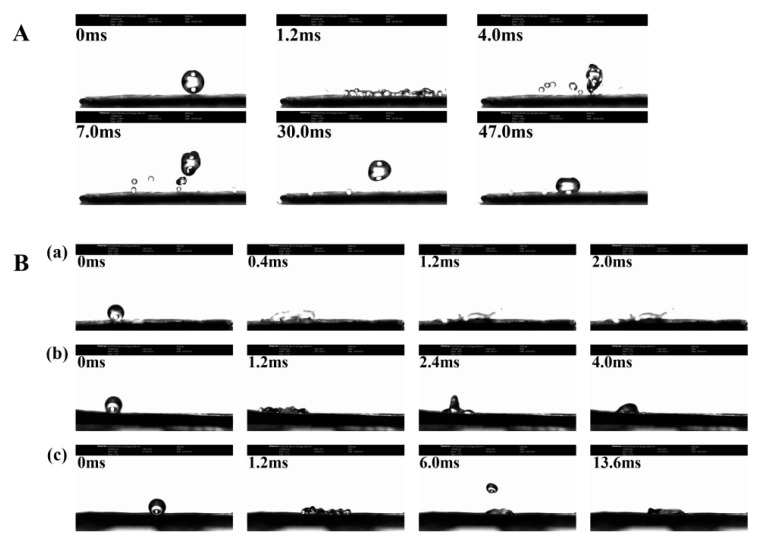
Bounce behavior of different samples on the surface of cabbage leaves. (**A**) Water droplets; (**B**) NCT/AOT vesicle under different dilution factors: diluted (**a**) 200 times, (**b**) 500 times, and (**c**) 1000 times.

**Figure 7 molecules-27-06916-f007:**
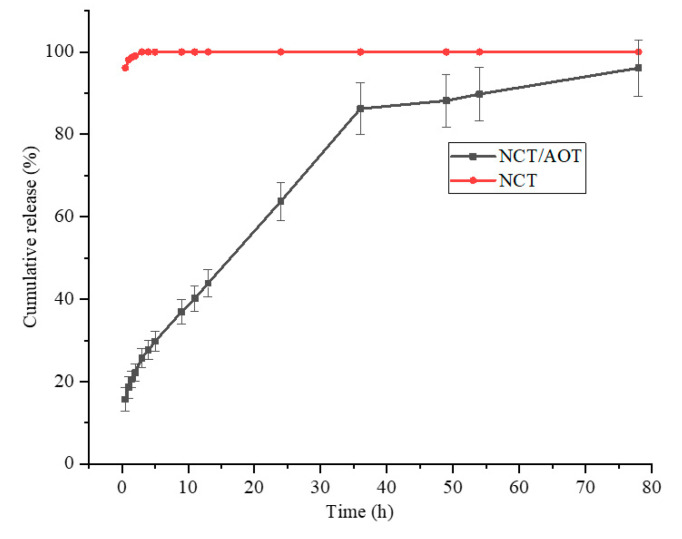
Release behavior of NCT from NCT/AOT vesicle and NCT aqueous solution.

## Data Availability

Not applicable.

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
