# Peer review of "Enhanced Insecticidal Effect and Interface Behavior of Nicotine Hydrochloride Solution by a Vesicle Surfactant"

_molecules, 2022, doi:10.3390/molecules27206916_

Round 1

Reviewer 1 Report

Comments about manuscript: molecules-1927283 titled “Enhanced insecticidal effect and interface behavior of Nicotine hydrochloride solution by a vesicle surfactant”.

The manuscript presents very interesting and actual problem connecting with the practical application of pesticides, especially in relation to the replacement of the synthetic and toxic compounds by those which are more environmentally friendly. One of such compound is Nicotine. However there are not too much information in the literature treated about the interfacial properties of Nicotine and its usage as a pesticide. Authors noticed that it is mainly connected with the low solubility of Nicotine in water. Because of that it is applied in some other different systems. Authors noticed that it is possible to increase Nicotine solubility in water by introducing it in vesicles of studied anionic surfactant (AOT). Presented results are very nice however the masurements and results of contact angle are discussive. Especially the cabbage leaf surface is not soft and homogeneous, and in such a situation the Young equation should not be used. The characteristic of the studied surface should be analysed, especially that the contact angle pictures show that surface effects could be the reason of not true contact angle values. In addition the statistical analysis of the presented results is should be described, the standard deviations are very small taking into account the rough surface of cabbage leaf (Fig. 4). There is no information about the liquid drop volume which was put on the leaf surface. The preparation of the surface is also important (especially eventual  cleaning). The contact angle values could be very valuable but only in the case when the measurement was done in the proper way. Comments should be improved because if the contact angle is about 30 degrees it means that there is no complete wetting. Is it good in the case of pesticide application?

In addition the analysis of the contact angle should be done for the AOT alone to compare the obtained results. Despite of my comments the manuscript are well organized and presented so I recommend it to the publication in Molecules however after the full explanation step by step the procedure of the contact angle  measurement and analysis of the surface topography of cabbage leaf.

Reviewer 2 Report

Review Ms: molecules-1927283

The manuscript entitled “Enhanced insecticidal effect and interface behavior of Nicotine hydrochloride solution by a vesicle surfactant” by Xiao et al., reports the study of vesicle solution formed by the surfactant AOT and nicotine. The authors used several techniques such as DLS, surface tension, SAXS and cryo-TEM.

In my opinion, the paper needs major revisions as noted to consider for publication. The general idea is interesting and could provide basic information for a continued effort find suitable systems for agricultural applications. However, there are several critical aspects of their interpretation that are not adequately justified or explained as I will show below so, in order to improve the discussion presented I suggest several comments.

Comments:

1.      My major criticism is that the work is showed in a confuse way and the reading is hard. For example, is not clear which ratio between AOT/nicotine is used in several part of the work. I suggest to put clearly in every caption. Additionally, the sample preparation should be informed in the first part of the results. Particularly making reference in the ClNa remotion. For example, in section 2.1. is not clear if the counterions are present or not. Please clarify.

2.      It is quite confused the experiments about encapsulation efficiency and pesticide loading of NCT/AOT sample. If the mixture is 1:1 is not necessary to calculate, but I am not sure if is correct my interpretation. Please clarify.

3.      In the DLS section, I have some questions: i) What was the analysis method used for DLS experiments: contin, cumulant, etc? ii) the data presented is by intensity, number, volume or Z average?

4.      The authors invoke the vesicles formation using SAXS, but it is not clear if the vesicles are unilamellar o multilamellar?

5.      I am not agreeing with the idea that if the ratio of nicotine hydrochloride increased, the SO3- group is protonated. Have the authors determined the pH of the aqueous solutions to support that idea?

6.      With regard to the release profile, how the authors explain the change at 36 hs?

Round 2

Reviewer 2 Report

Review Ms: molecules-1927283 v2

Even that the new version of the MS is improved, I have some minimal comments to consider about the DLS data. In particular, Z average is not appropriated to describe the system formed. Indeed, Figure 3B cryo-TEM image show several populations that cannot be described by the Z average value. I suggest to present the DLS data by intensity. See similar works about that such as: Langmuir, 2019, 35, 13332–13339; Langmuir, 2020, 36, 10785–10793; RSC Adv., 2017, 7, 5372–5380; Chem. - A Eur. J., 2012, 18, 15598–15601; Phys. Chem. Chem. Phys., 2015, 17, 17112–17121; Colloids Surfaces A Physicochem. Eng. Asp., 2020, 606, 125435; RSC Adv., 2018, 8, 12535–12539. Additionally, several authors have invoked that AOT can form different organized media (direct micelles, worm like micelles and vesicles) depending the counterions and concentration. Thus, the novel system described by the authors should mentioned the possibility to coexist micelles and vesicles.

Author Response

The DLS was re-mearsured and intensity was shown in the modified version.  It was found that the size of DLS and the value of PDI decreased with the increase of dilution ratio, which may be because some vesicles transformed into micelles and formed a vesicle-micellar coexistence system.